

# Variational quantum classifier-based early identification and classification of chronic kidney disease using sparse autoencoder and LASSO shrinkage

P. Parthasarathi[1], Haya Mesfer Alshahrani[2], K. Venkatachalam[3] and Jaehyuk Cho[4]

[1] Department of Computer Science and Engineering, Bannari Amman Institute of Technology, Erode, India
[2] Department of Information Systems, College of Computer and Information Sciences, Princess Nourah bint Abdulrahman University, Saudi Arabia
[3] Department of Software Engineering, Jeonbuk National University, Jeonju, Republic of Korea
[4] Department of Software Engineering and Division of Electronics Information Engineering, Jeonbuk National University, Jeonju, Republic of Korea

Corresponding author
Jaehyuk Cho, chojh@jbnu.ac.kr

## ABSTRACT

The two leading causes of chronic kidney disease (CKD) are excessive blood pressure and diabetes. Researchers worldwide utilize the rate of globular filtration and kidney inflammation biomarkers to identify chronic kidney disease that gradually reduces renal function. The mortality rate for CKD is high, and thus, a person with this illness is more likely to pass away at a younger age. Healthcare professionals must diagnose the various illnesses connected to this deadly disease as promptly as possible to lighten the impact of CKD. A quantum machine learning (QML) based technique is presented in this research to help with the early diagnosis and prognosis of CKD. The proposed research comprises four phases: data pre-processing, data augmentation, feature selection, and classification. In the first phase, Kalman filter and data normalization techniques are applied to handle the missing and noisy data. In the second phase, data augmentation uses sparse autoencoders to balance the data for smaller classes. In the third phase, LASSO shrinkage is used to select the significant features in the dataset. Variational Quantum classifiers, a supervised QML technique, are employed in the classification phase to classify chronic kidney diseases. The proposed system has been evaluated on the UCI dataset, which comprises 400 CKD patients in the early stages with 25 attributes. The suggested system was assessed using F1-score, precision, recall, and accuracy as evaluation metrics. With a 99.2% classification accuracy, it was found that this model performed better than the other traditional classifiers used for chronic kidney disease classification.

# INTRODUCTION

As chronic kidney disease has a high death rate, it has drawn a lot of interest in recent times. The World Health Organisation (WHO) states that these kinds of chronic illnesses

are becoming a threat to emerging nations. Around the world, chronic kidney failure claimed the lives of 786 million people in 2018 (*Shih et al., 2020*). Of those, 354 million were men, and 432 million were women. Kidney illness is referred to as "chronic" because it impairs the ability of the urine system to operate and develops progressively and extends for an extended period (*Qin et al., 2019*). Waste materials build up in the bloodstream and cause many health issues, including diabetic complications, elevated or decreased pressure levels, damage to the nervous system, and disorders of the bones that ultimately result in coronary artery disease (*De Almeida et al., 2020*).

Presently, approximately 15% of people worldwide are afflicted with chronic kidney disease, an excruciating and eventually fatal illness. This illness is made even more deadly by the fact that it cannot be identified until significant kidney damage has already occurred (*Shankar et al., 2020*). When a patient discovers they have the ailment, it becomes a tedious and drawn-out process to get them tested, diagnose a potentially incorrect result, prescribe medication based on the stage of chronic Kidney failure they may be in, and provide all the care necessary to keep them alive (*Deepika et al., 2020*).

Patients with kidney diseases are at risk for hypertension, obesity, and heart failure. Especially in the more advanced stages of the disease, these patients experience adverse effects that impair their immunological and cognitive systems. Patients may progress to the point where they require hemodialysis or a kidney replacement in underdeveloped nations (*Ma et al., 2020*). Glomerular screening level, a measure of kidney function, is used by healthcare providers to diagnose kidney disease. This measure is determined based on variables like sex, lifespan, and outcomes of blood tests. Healthcare providers can categorize this kidney disease into five phases based on the glomerular screening level value: standard, mild, moderate, severe, and kidney failure (*Haq et al., 2020*).

Chronic kidney disease is most commonly detected by using a bloodstream molecular composition and urinalysis or by finding the medical condition as a side effect of an additional treatment. Less common manifestations include diminished generation of urine, urinary tract infections, abdominal apprehension, excessive hemorrhages, and frothy urine (*Sabanayagam et al., 2020*). People with severe chronic kidney failure have symptoms such as exhaustion, decreased desire to eat, dizziness, vomiting, rusty flavor, unanticipated shedding of pounds, irritation, modifications to their psychological state, breathing problems, or regional swelling (*Peng et al., 2021*).

Kidney failure can be mitigated by treating persistent kidney disease and discovering it promptly. Two diagnostic procedures are utilized in determining the presence of chronic kidney disease: an examination of the blood to measure albumin or an examination of the urinary tract to measure the filtrate produced by the kidneys (*Kumar, Sinha & Bhardwaj, 2020*). Computer-aided examinations are needed to support the clinical decisions by by physicians and practitioners of the rising number of individuals with with chronic kidney problems, the lack of expert medical professionals, and the expensive nature of treatment and procedures, particularly in countries with limited resources (*Daniel et al., 2021*).

The lack of a single, broadly helpful indicator that can distinguish between healthy and sick individuals is a significant factor in chronic kidney disease and chronic illnesses around the globe. This makes it more difficult for doctors and researchers to quickly and accurately diagnose this illness, resulting in inaccurate predictions of illness (*Parab et al., 2021*). The machine learning models that are supervised in nature are capable of performing categorization and learning different data patterns. To increase test accuracy, a robust classification model unaffected by changing circumstances is needed (*Rashed-Al-Mahfuz et al., 2021*). With proper adjustments to the parametric variables and sufficient input data, the neural network algorithms can distinguish those with chronic kidney disease from other normal individuals with high precision during testing (*Roth et al., 2021*).

Neural networks have been demonstrated to be beneficial in the administration of medication to patients with chronic kidney failure. Neural networks are an excellent fit for diagnosing chronic kidney disease since they have become increasingly proficient in categorizing, forecasting, connecting, *etc*. (*Jeong et al., 2020*). Massive clinical data from practical applications is the foundation upon which medical artificial intelligence has been developed. It is difficult for individuals to directly study large data sets due to the time and attention required to prevent human error and the need to extract the insights or information in detail (*Xin et al., 2020*). It is evident that in some categories, artificial intelligence systems outperform humans by orders of magnitude. Research on quantum machine learning (QML) techniques for kidney illness is only getting started. The main areas of focus for current research on the role of quantum systems in kidney disease are prediction evaluation, planning for therapy, notification systems, and testing aids (*Yuan et al., 2020*). This research aims to implement transfer learning strategies to obtain a high-level accuracy classification on the NIH chest X-ray dataset, which includes images with diseases of the chest, such as pulmonary lesions (*Shamrat et al., 2023*; *Ghosh et al., 2020*). The research demonstrates how machine learning models can detect risk factors for heart disease, which is an important issue. This seems to be because machine learning can process large amounts of data and find patterns often beyond human capabilities (*Ghosh et al., 2021*).

Kidney disease is a significant worldwide medical and public health burden due to its high rates of hospitalization and premature death, as well as its substantial financial costs associated with both short- and long-term kidney disease (*Senan et al., 2021*). Patients with kidney disease exhibit significant variations in the look, course, and responsiveness to therapy of their condition. Quantum-based techniques can help clarify targeted treatment for more precise morphological and predictive results in kidney sickness.

Feature selection is an essential step in disease prediction and classification. An automatic filtering method is a data analysis and feature selection approach that does not rely on a classification algorithm (*Kanda, Kanno & Katsukawa, 2019*). While the learning algorithm is occupied with other duties, this technique can remove unnecessary data points from a dataset without involving the learning algorithm. Thus, the proposed work

intends to employ a supervised QML technique for chronic kidney disease prediction, deep learning-based data augmentation, and feature selection techniques.

### Research contributions

The main contributions of this research are,

1) To leverage the variational quantum classifier to classify chronic kidney diseases to enhance the prediction accuracy and efficacy.
2) Employ data augmentation techniques such as sparse autoencoder and feature selection techniques such as LASSO shrinkage to mitigate the issue of multicollinearity in data.
3) To assess the performance of the proposed system and compare it with the existing methods to demonstrate the significance of the suggested quantum-based classification method.

### Paper organization

The remainder of the article is organized as follows. "Related Works" presents the current works using machine learning and deep learning in the literature concerning chronic kidney disease prediction. "Proposed Methodology" describes the proposed methodology, which includes stages such as preprocessing, augmentation, feature selection, and classification. "Results and Discussion" discusses the outcomes of the experimental assessment of the proposed system on the CKD dataset. "Conclusion" concludes the present research.

## RELATED WORKS

This section elaborates on the recent developments using machine learning and deep learning algorithms for detecting and categorizing kidney disease. The authors proposed the successive limit proximity and pixelwise categorization networks in *Song et al. (2020)* for independent kidney delineation from ultrasound imagery. The investigators initially described using previously trained neural networks with deep learning to classify significant images from ultrasound scans. They then used these functions as input for a secondary separation statistical network to learn kidney border visuals. Finally, they used an algorithm to classify pixels and designate the identified border separation links into the kidney pixels. To forecast the kidney and kidney borders in the conclusive learning technique, the kidney imaging segmentation is based on sophisticated convolutional neural networks.

Prospective authors in another work used MATLAB to analyze the surface properties of ultrasound kidney images. Statistical processes were then applied to those images to establish a distinction between the kidneys that were afflicted and those that were healthy. Among the statistical procedures carried out, it was determined that the square root of the average over the whole kidney area and the region of the cortex produced the best categorization results. The 93% accuracy rate of the naive Bayes approach for kidney disease detection reported in *Belur Nagaraj et al. (2020)* demonstrates that machine learning methods performed adequately when applied to kidney datasets.

Patients with chronic kidney disease frequently experience hemophilia, which increases their susceptibility to the spread of infection. Delivering drugs in optimal condition and as soon as feasible is imperative. Repetitive region framework intelligent control, which also controls erythromycin, is a strategy described in *Zhang et al. (2021)* and may be used to manage iron levels in patients with kidney diseases. In *Segal et al. (2020)*, it was discovered that boosting techniques, such as gradient boosting, extreme gradient boosting (XGBoost), XGBoost and ID3 decision tree, helped assess the precision of kidney disease categorization. In *Krishnamurthy et al. (2021)*, it was demonstrated that Random Forest was a more accurate kidney illness predictor than k-nearest neighbour and logistic regression classifiers.

The intelligent fuzzy reasoning network was developed by *Shang et al. (2021)* to forecast prolonged kidney damage. Here, the number of uncertain guidelines and the associative functions of the initial variables coincide. Surface screening rates have been identified, and the system's reliability has been compared with other neural network methods. The models based on the fuzzy technique for determining exceptionally reliable screening rates are demonstrated by modeling the predictive results from the proposed networks. The primary concerns of the clinicians include how well those with chronic kidney disease are being treated, how often they are being monitored, and how to slow down the development of the disease and avoid its eventual repercussions. A meaningful way to increase the burden of this research is to use the reliable forecasting system that was started for quantitative decision support about kidney conditions and their corresponding therapy.

According to the authors, the structure identification method was first implemented in chronic kidney failure using the previously established revamped shared data (*Schena et al., 2021*). This study presents an updated version of interrelated data that allows one to distinguish between beneficial and detrimental relationships. Clinicians extensively confirmed all sixteen relevant patterns the researchers revealed about the illness. They used collaborative data to demonstrate the class relationship rules to the suggested schema to reduce computational overhead. Using illness data, algorithmic evaluation was conducted, and all the identified sixteen patterns were clinically generated. It could help in identifying genetic markers that predict an individual's response to certain treatments or their likelihood of developing a disease (*Li et al., 2024*; *Huang et al., 2022*). Super-resolution refers to the process of enhancing the resolution of an image, improving its clarity and detail beyond the original capture quality (*Jia, Chen & Chi, 2024*; *Song et al., 2024*). Enhanced images can aid in monitoring the progression of retinal diseases or the effects of treatments (*Fan et al., 2024*).

The authors presented a novel method for applying transfer learning based on neural network architectures such as DenseNet, ResNet, and Squeeze Net (*Weber et al., 2020*) and compared them for detecting those with neurological disorders. All employed visuals had been previously processed using logarithmic lengthening, and the three layers were given unique cumulative coefficients for learning. The experimental results demonstrated that, when compared to the most recent techniques, the suggested DenseNet model is the best.

*Jeong et al. (2021)* created a model for evaluating intracerebral perforation based on the Mask R-CNN segmentation technique. The model uses Kalman reduction to eliminate

**Table 1 Comparison of state-of-the-art methods on chronic kidney disease.**

| References | Techniques | Inferences | Performance |
| --- | --- | --- | --- |
| Haq et al. (2020) | Support Vector Machine | Limited number of data samples were used | Accuracy = 85.2% |
| Sabanayagam et al. (2020) | Artificial Neural Network | Feature selection is not performed | Accuracy = 90% |
| Kumar, Sinha & Bhardwaj (2020) | Ensemble learning | High training time | Accuracy = 92.6% |
| Daniel et al. (2021) | 11 classifiers | High accuracy is achieved | Accuracy = 95.9% |
| Rashed-Al-Mahfuz et al. (2021) | Pre-trained CNN architectures | System design is complex | Accuracy = 91.4% |
| Roth et al. (2021) | Deep Neural Network with Artificial Bee Colony Optimization | Large and diverse dataset is used | Accuracy = 92.6% |
| Xin et al. (2020) | AdaBoost, Logistic Regression | Decision support system for kidney disease prediction is proposed | Accuracy = 93.5% |
| Yuan et al. (2020) | Radial Basis Function Networks | High values of training and validation loss is achieved | Accuracy = 94.8% |

distortion and improve the visual appearance, as well as sophisticated deep learning for features to be extracted. Using the CNN classifier for prognosis and the multilayer autoencoder model to extract the most efficient and valuable characteristics from the chronic kidney disease dataset, *Shih et al. (2020)* developed a deep learning categorization architecture that yielded a high precision of 95.3%. Imaging specialists can use the transfer learning-based VGG-16 model to identify kidney disorders and aid in patient diagnosis. Using data enhancement techniques such as random interpretation, brightness rectification, chaos administration, expansion, and image whirling—VGG16 was employed as the fundamental transfer learning system to enhance the efficacy of classification. The prediction model developed and verified by *Dovgan et al. (2020)* with data from the neighborhood healthcare chain was based on supervised machine learning algorithms. Inference from the Isolation Forest method was used to build the framework, and indicators of the degree of statistical fitting were used to assess it. Table 1 provides a comparison of various chronic kidney disease methods.

## Research gaps and motivations of current research

Current CKD diagnosis techniques rely on conventional machine learning models, which have trouble with noisy medical data and class imbalance, resulting in less-than-ideal accuracy and delayed detection. Quantum computing presents a promising solution for handling big datasets more effectively and efficiently because it can represent and analyse data in significantly larger spaces. Furthermore, the majority of studies do not use quantum machine learning (QML) to increase classification performance or sophisticated data augmentation strategies to solve class imbalance. This study investigates a QML-based strategy with improved feature selection, data preprocessing, and classification methods to close these gaps and increase the precision and resilience of early CKD diagnosis. Compared to conventional models, quantum models such as VQCs can more thoroughly

investigate an ample solution space by taking advantage of quantum superposition and entanglement.

# PROPOSED METHODOLOGY

The proposed methodology involves four different phases for the classification of chronic kidney disease. The various phases in the proposed system include preprocessing, data augmentation, feature selection, and classification, as shown in Fig. 1. Each of these phases and their techniques are elaborated in detail in this section.

## Data preprocessing

Preparing the data is the most important stage before using classification algorithms. The prediction task cannot directly use real-world data due to its high noise, incompleteness, and inconsistency. A preprocessing process is applied to appropriately represent the data for predicting chronic kidney disease. Handling missing data and data normalization is part of this research's data preprocessing procedure.

### Kalman filter

Two fundamental presumptions underlie the Kalman filter's operation. First, the data is a linear dynamic system, with the current state inferred from observations and past states.

Secondly, random measurement fluctuations follow a normal distribution because the noise is Gaussian-distributed. The suitability of this technique to present research is supported by the fact that the renal function biomarkers in CKD datasets show gradual, smooth changes over time, which is consistent with the dynamic model of the Kalman filter. Additionally, Gaussian noise can occur in medical measures because of environmental influences, human error in input, or equipment sensitivity. The selection of Kalman filtering was validated by exploratory data analysis, which showed that missing values and variations in experiment outcomes were following Gaussian noise. Since the Kalman filter can handle sequential and continuous data, like the time-series biochemical indicators seen in the CKD datasets, it was selected above alternative imputation and noise reduction techniques. Instead of making imprecise assumptions about missing data, the Kalman filter efficiently estimates missing values by utilizing estimates from the past and present states. The Kalman filter produces more accurate reconstructions of missing or noisy values than static imputation techniques because it dynamically changes predictions based on trends found in patient records.

Noise and missing values cause the prediction to be less accurate or produce a result that is not reliable. This filter purges the data by eliminating noise, redundant information, and contradictions. In the data filtering stage, two more independent filters are used to substitute missing values and eliminate ineffectual data. The initial filter uses the median and the average of the available data to fill in all missing values in the formatted dataset. The subsequent filter eliminates meaningless attributes with a variance of ninety percent

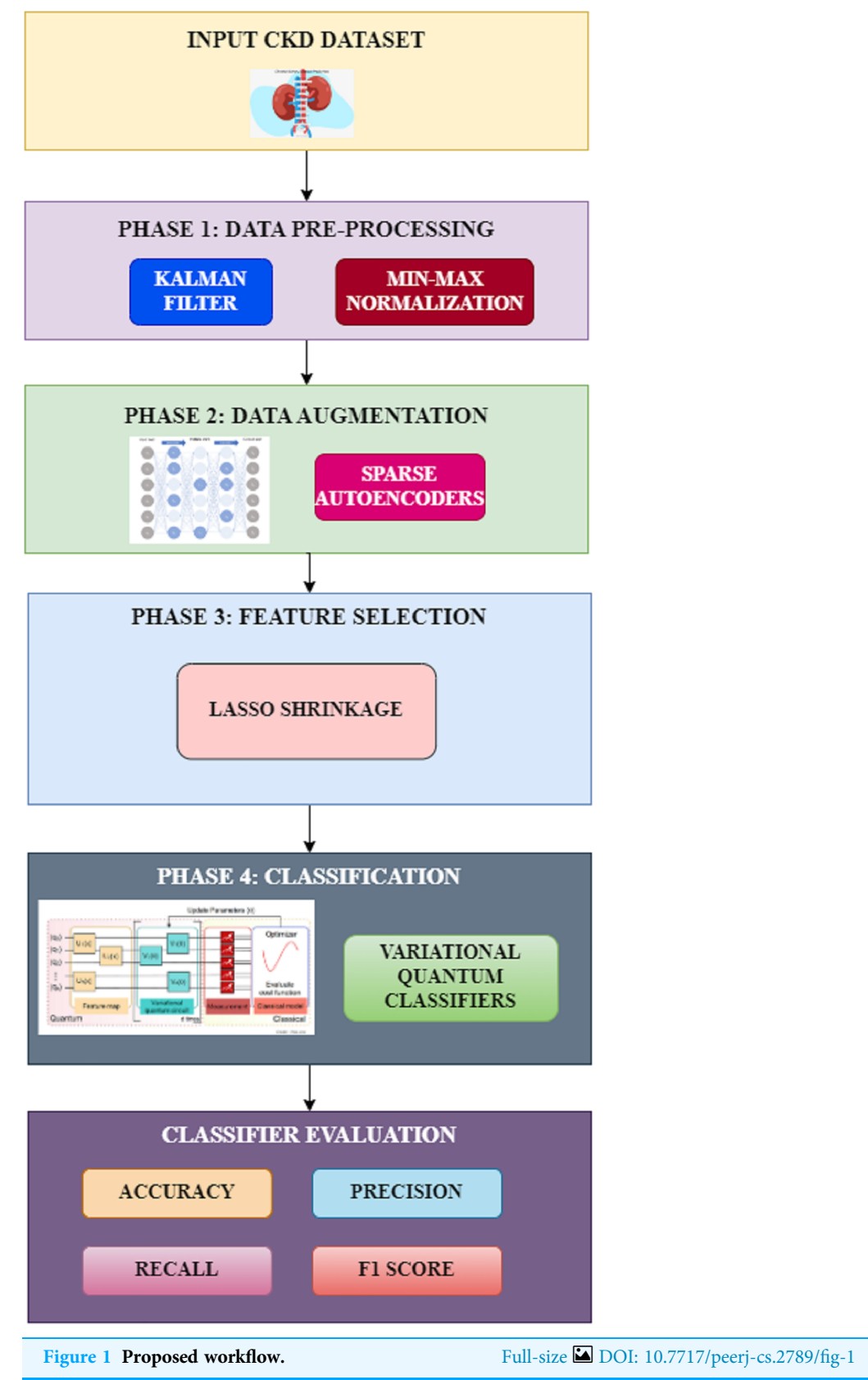

**Figure 1  Proposed workflow.**               

and above. The mathematical formulation for handling missing values is as given in Eq. (1):

$$\bar{H} = \frac{1}{K} \sum H_m^{J_n}.$$  (1)

In this equation, $H$ represents the features in the dataset, $J$ corresponds to the category mapping of each feature, and $\bar{H}$ denotes the average value of the features.

### Data normalization

There are many features in the chronic kidney disease dataset, and each feature has a distinct collection of numerical information, making computation more complicated. Therefore, to reduce the numerical intricacy during the computational procedure of kidney disease prediction, the data normalization technique is utilised to normalise the dataset in the range between zero and one. The min–max normalization technique, as represented in Eq. (2), is applied in the suggested system. In Eq. (2), $S_{norm}$ denotes the normalised form of the data value, $S$ represents the value as per the dataset, $S_{max}$ and $S_{min}$ corresponds to the maximum and minimum values in the dataset. The updated maximum and minimum values are denoted as $upd_{max}$ and $upd_{min,}$ respectively.

$$S_{norm} = \frac{S - S_{min}}{S_{max} - S_{min}} \times [upd_{max} - upd_{min}] + upd_{min}.$$  (2)

## Data augmentation

When analyzing a dataset that comprises labels for all the features, it is common to observe that certain classifications outnumber the rest of the data. This is a regular occurrence within healthcare datasets, where samples with special conditions are used to make up a smaller fraction than normal samples. This may result in classification algorithms considering only the particularly congested classes, failing to pay attention to these smaller classes. Sparse autoencoders are utilized in this study for data augmentation to address this problem.

### Sparse autoencoder

The primary characteristic of this autoencoder is that rather than just representing each sample, the encoder learns the typical distribution of the input data. The hidden layer samples a new element using the recently trained and typical distribution. After the autoencoder is trained, it will produce additional elements of the dataset that fit into the same typical distribution of original data whenever an additional data component is added to the network. It is possible to create synthetic information comparable to the actual one using this approximation type. The loss function needs to be modified to map the synthetic data with the typical data distribution, which uses the KL divergence technique as shown in Eq. (3):

$$SP_{loss} = ||a - \bar{a}||^2 + KL\,[T(\mu_a, \sigma_a),\ T(0, 1)].$$  (3)

In Eq. (3), $\bar{a}$ denotes the newly constructed data, $T(\mu_a, \sigma_a)$ represents the typical data distribution with average as well as standard deviation values such as $\mu_a$ and $\sigma_a$ respectively. The value of KL divergence is computed as per the formulation shown in Eq. (4):

$$KL[i,j] = -\int i(a)\log j(a)\ da\ +\ \int i(a)\log i(a)\ da. \tag{4}$$

L1 regularization method was introduced in the network to make the network utilize only a minimal number of neurons during the training process. Hence, the weight parameter is added to the loss function and represented as in Eq. (5),

$$SP_{loss} = Err(u,\ \hat{u}) + \gamma \sum_{k=1}^{N} W_k. \tag{5}$$

In the above equation, $W_k$ denotes the weight corresponding to $k$th neuron, $\gamma$ denotes the multiplier parameter, $u$ and $\hat{u}$ represents the actual and predicted data values. A larger multiplier factor indicates a more significant impact of regularisation over the entire loss computation. As a result, the network can describe the original data with additional attributes, enabling it to examine the data from an alternative viewpoint.

## Feature selection

Numerous features that might not be pertinent to the intended prediction can be found within healthcare datasets. To focus the model on the most significant characteristics and reduce the overall dimension of the data, feature selection approaches are especially helpful. Feature selection aids in the construction of a more precise and effective model by choosing the most pertinent features. It lowers the interference from superfluous or extraneous characteristics, which could have a harmful effect on the model's functionality. This is especially crucial for the categorization of CKD since the presence of extraneous variables can result in overfitting issues or incorrect forecasts. This study uses the Least Absolute Shrinkage and Selection operator (LASSO) to select the features.

### LASSO shrinkage

This technique is useful for choosing feature variables and enhancing the resulting model's comprehension and reliability for prediction. The LASSO method aids in choosing features and the removal of variables by condensing the data values to a single spot. This kind of technique works well with extremely overlapping or interrelated models. The LASSO shrinkage adds an adjustment proportional to the absolute value of the coefficient magnitudes; from that, some coefficients are finally removed from the model after becoming zero, leaving a model with fewer coefficients because of variable removal. This technique is aimed towards minimizing the formulation as represented in Eq. (6) as

$$L_S = \sum_{m=1}^{N} \left( u_m - \sum_{m} a_{mx}\delta_x \right)^2 + \beta \sum_{x=m}^{d} |\delta_x|. \tag{6}$$

The parameter $\beta$ in the above equation denotes the factor used for modifying the level of shrinkage. After the shrinkage procedure, a particular group of values ($\delta$) becomes zero. When $\beta = 0$, no variables are eliminated from the model. As $\beta$ increases, more parameters are removed from the model and set to zero. A decrease in $\beta$ increases variability, while an increase in $\beta$ elevates bias values. The $\delta$ value for a parameter indicates the significance of that variable concerning its influence on the substrate variance. When $\delta = 0$, a variable is deemed insignificant and disregarded. It should be mentioned that LASSO regression produces deceptive findings when there is instability in the dataset, which could lead to choosing inappropriate and crucial variables when LASSO is applied to the entire dataset. If a method is used that runs the LASSO repeatedly and arbitrarily selects subgroups from the dataset, the impact of instability in data will be lessened. In most iterations, majority voting chooses the higher than zero variables based on $\delta$ values.

## Classification

Classification is the problem of identifying the class label for an unknown data point for T number of total classes. For any given dataset with labels, the classifier is mathematically formulated as in Eq. (7):

$$P = \{(a_1, u_1), \ldots, (a_N, u_N)\}\ H^N \times \{0, 1, \ldots, T-1\}. \tag{7}$$

In this research, the variational quantum classifier which is a supervised quantum machine learning algorithm is trained to reduce the cost function through the optimization of the quantum gates as shown in Eq. (8),

$$M(\omega) = \sum_{k=1}^{N} \theta_k m(u_k, h(a_k, \omega)). \tag{8}$$

In the above equation, $h(a_k, \omega)$ denotes the model trained to identify the label of the input $a_k$ and the corresponding output is represented by $u_k$.

### Variational quantum classifier

Several advantages are associated with using quantum models for CKD classification compared to classical machine learning (ML) models. Complex, high-dimensional medical data, such as CKD biomarkers, can be better represented because of quantum states' exponentially huge feature spaces. While quantum models use quantum entanglement and superposition to map features to an augmented space where patterns are more straightforward to distinguish, classical ML models use linear or polynomial kernels. Compared to ML models, variational quantum classifiers (VQCs) optimise fewer parameters, lessening the computational load. While quantum circuits use only a few

tuneable parameters with tremendous generalisation potential, classical deep-learning models require millions of parameters.

A VQC is one important QML technique for differentiating physical events of interest from background events. For classification issues in the chaotic intermediate-scale quantum computing device, it is a popular supervised QML approach. The exploratory results on quantum devices can be obtained using this method, which eliminates the requirement of further strategies for rectification of errors. Based on quantum circuits that are challenging to replicate conventionally, this quantum technique maps conventional input data to an increasingly large quantum spectrum of features. In this classifier, conventional data is embedded into quantum computing through various feature mapping and encoding approaches, beginning with the state preparation. Ultimately, the measurement result is fed back into a circuit to refine the parameters that can be trained in the variational circuit. There are two stages associated with the VQC algorithm such as training and testing.

*State preparation*

This step is essential when applying quantum machine learning algorithms to process data. The traditional machine learning model that performs two-class classification on a dataset is represented as shown in Eq. (9):

$$C = \{(a_1, u_1), \ldots (a_m, u_m), \ldots (a_N, u_N)\}. \tag{9}$$

In the above equation, $a_m$ denotes the features of the sample $m$ and $u_m$ denotes the corresponding output prediction for the target class labels represented as $L = \{cl_1, \ cl_2, \ldots cl_n\}$. For two-class classification problem, $u_m \ \varepsilon \ \{cl_1, \ cl_2\}$. To investigate the data in the context of quantum machine learning algorithms, it is necessary to convert the conventional data into quantum format $(|\psi_m\rangle)$ as represented in Eq. (10),

$$Q_m = \{(|\psi_1\rangle, u_1), \ldots, (|\psi_m\rangle, u_m), \ldots, (|\psi_N\rangle, u_N)\}. \tag{10}$$

*Hybrid data encoding*

High-dimensional quantum data can be embedded into conventional data using various methods. In this proposed research, a hybrid encoding technique is used, combining amplitude and qubit encoding techniques. When the quantum circuit width is taken into account, the amplitude encoding is favourable, and when the quantum circuit depth is taken into account, the qubit encoding is beneficial. The extremes of quantum circuit complexity for loading conventional data into a quantum system are represented by these two encoding approaches. A hybrid encoding technique is included in the suggested system to reduce the quantum circuit complexity between these two extremes. The hybrid encoding simultaneously applies amplitude encoding to several separate qubit units.

Assume $h$ to be the number of qubit units in each block $D$ that is the encoded form of conventional data created using amplitude encoding. The resultant quantum system will

constitute conventional data in the order of $D2^h$. This encoded data is represented as shown in Eq. (11) as

$$V_\theta(a) : a\varepsilon R^N \rightarrow |\theta(a)\rangle = \otimes_{s=1}^D \left( \frac{1}{||a||_s} \sum_{t=1}^{2^h} a_{st}|s\rangle_t \right). \tag{11}$$

As every segment may have a distinct normalization constant, amplitudes may not accurately reflect the data unless the parameters of the normalization constant are similar. Hence, the qubit encoding scheme is introduced in this stage to tackle this issue. The modified quantum representation is as shown in Eq. (12):

$$|\theta(a)\rangle = \otimes_{r=1}^D \left( \sum_{s=1}^{2^h} \prod_{t=0}^{h-1} cos^{1-s_t}(a_{f(t),r})sin^{s_t}(a_{f(t),r})|s\rangle_t \right). \tag{12}$$

In the above equation, $s \in \{0,1\}^m$ is the two-coded representation of $s$ and $s_t$ denotes the $t + 1^{th}$ unit of the data bit. $x_{s,t}$ denotes the $s^{th}$ unit of the data bit. $x_{s,t}$ denotes the $s^{th}$ data unit assigned to $t^{th}$ block of qubits. Compared to qubit encoding, hybrid encoding techniques employ fewer qubits and a shorter quantum circuit depth than amplitude encoding.

*Feature mapping*

A quantum feature map encodes conventional data in the realm of the quantum field using a quantum network built from the traditional machine learning kernel technique. The data are projected into a higher-dimensional Hilbert space to identify a distinct hyperplane for the classification of data that is not linear. The data is transformed using unitary gates as represented in Eqs. (13) and (14):

$$v_\theta(a) = V_{\theta(a)}HD^{\otimes k}V_{\theta(a)}HD^{\otimes k} \tag{13}$$

$$V_{\theta(a)} = \exp\left( k \sum_{F \subseteq [k]} \theta f(a) \prod_{k \in F} D_k \right). \tag{14}$$

In Eq. (13), $HD$ corresponds to the Hadamard gate, the unitary gate is denoted using $V_{\theta(a)}$ in the context of the Pauli feature method, and $D_k$ represents the constructed feature space. The number of qubits required depends on the quantity of the data, and unitary gates are used to describe the data by varying the angle to specific levels.

*Variational circuit*

This method's fundamental principle is to optimize the parameter values by following the directions of a value function. The quantum phase in the variational quantum classifier comprises state preparation, measurement, and the parameterized input $a$ of the the circuit, which depends on the total quantity of parameters such as $R_y$, $R_z$ and *CNOT gate*. The classical phase comprises the training method, the objective function, and the circuit's

output. The VQC is estimated *via* optimization approaches, such as restricted optimization by linear estimates. The problem solved by this circuit is represented as shown in Eq. (15):

$$|\psi(a:\varphi)\rangle = V(\varphi)|\delta(a)\rangle. \tag{15}$$

*Measurement*

The aim of training the model is to find the appropriate values for parameters that will optimise a specific loss function. Similar to how a traditional neural network is optimised, a quantum model can also be optimised. The model is executed in a forward direction, and the loss function is found in both cases. Slope-based optimisation techniques can be employed to modify the parameters trained as a loss function. By using this technique, the difference between the truth and predicted outcomes can be calculated and represented by a loss function value as given in Eq. (16):

$$\langle Y \rangle_{|\delta\rangle} \equiv \langle \delta|Y|\delta\rangle = |\mu|^2 - |\sigma|^2. \tag{16}$$

In the above equation, $Y = \begin{bmatrix} 1 & 0 \\ 0 & -1 \end{bmatrix}$ and $\langle Y \rangle \in [-1, 1]$ for the actual value of output that is measurable.

*Optimization*

When the measurements are prepared, the quantum variational circuit's parameters are changed through an optimisation procedure. These parameters are trained using a conventional loop until the actual value of the cost function falls. Using a $n + 1$ principle (where $n$ is the number of features), the Nesterov Momentum Optimiser generates successive proportional estimates of the cost function and obstacles, improving these estimates at every step in a trusted region.

*Classifier evaluation*

The working conditions of the quantum-based circuit are represented in equations between Eqs. (17) and (20). These metrics are computed based on the true positive, false positive, false negative, and true negative values.

$$model_{prec} = \frac{True\ Positive}{True\ Positive + False\ Positive} \tag{17}$$

$$model_{acc} = \frac{True\ Positive + True\ Negative}{True\ Positive + False\ Positive + True\ Negative + False\ Negative} \tag{18}$$

$$model_{rec} = \frac{True\ Positive}{True\ Positive + False\ Negative} \tag{19}$$

$$model_{f1} = 2 * \frac{True\ Positive}{2\ (True\ Positive) + False\ Positive + False\ Negative}. \tag{20}$$

**Table 2 Dataset description.**

| S. No. | Feature variable | Feature description | Variable type | Value range |
|---|---|---|---|---|
| 1 | age | Age | Numerical | 0–90 |
| 2 | bp | Blood Pressure | Numerical | 0–180 |
| 3 | sg | Specific Gravity | Numerical | 0–1,025 |
| 4 | al | Albumin | Numerical | 0–5 |
| 5 | su | Sugar | Numerical | 0–5 |
| 6 | rbc | Red Blood Cells | Categorical | Normal/Abnormal |
| 7 | pc | Pus Cell | Categorical | Normal/Abnormal |
| 8 | pcc | Pus Cell Clumps | Categorical | Present/Not Present |
| 9 | ba | Bacteria | Categorical | Present/Not Present |
| 10 | bgr | Blood Glucose Random | Numerical | 0–490 |
| 11 | bu | Blood Urea | Numerical | 0–391 |
| 12 | sc | Serum Creatinine | Numerical | 0–76 |
| 13 | sod | Sodium | Numerical | 0–163 |
| 14 | pot | Potassium | Numerical | 0–47 |
| 15 | hemo | Haemoglobin | Numerical | 0–17.8 |
| 16 | pcv | Packed Cell Volume | Numerical | 0–54 |
| 17 | wbcc | White Blood Cell Count | Numerical | 0–26,400 |
| 18 | rbcc | Red Blood Cell Count | Numerical | 0–8 |
| 19 | htn | Hypertension | Categorical | Yes/No |
| 20 | dm | Diabetes Mellitus | Categorical | Yes/No |
| 21 | cad | Coronary Artery Disease | Categorical | Yes/No |
| 22 | appet | Appetite | Categorical | Poor/Good |
| 23 | pe | Pedal Edema | Categorical | Yes/No |
| 24 | ane | Anemia | Categorical | Yes/No |
| 25 | Class | CKD, not CKD | Categorical | CKD/not CKD |

## RESULTS AND DISCUSSION

This section discusses the dataset used to evaluate the proposed system's performance through experimental assessments and also describes the limitations of current research.

### Dataset description

The Chronic Kidney Disease dataset (*Rubini, 2015*) used in this research is obtained from the UCI Machine Learning repository. The dataset is composed of a total of 400 records for 25 features. The target variable in the dataset consists of two labels, CKD and not CKD. The remaining 24 variables can be employed as features to perform the prediction. The total number of samples for the CKD target category is 250, whereas the total number of records for the CKD category is 150. A brief description of the variables available in the dataset is given in Table 2. The distribution analysis of numerical and categorical features in the CKD dataset is presented in Figs. 2 and 3. The dataset used in this research can be

Distributions of numerical Features

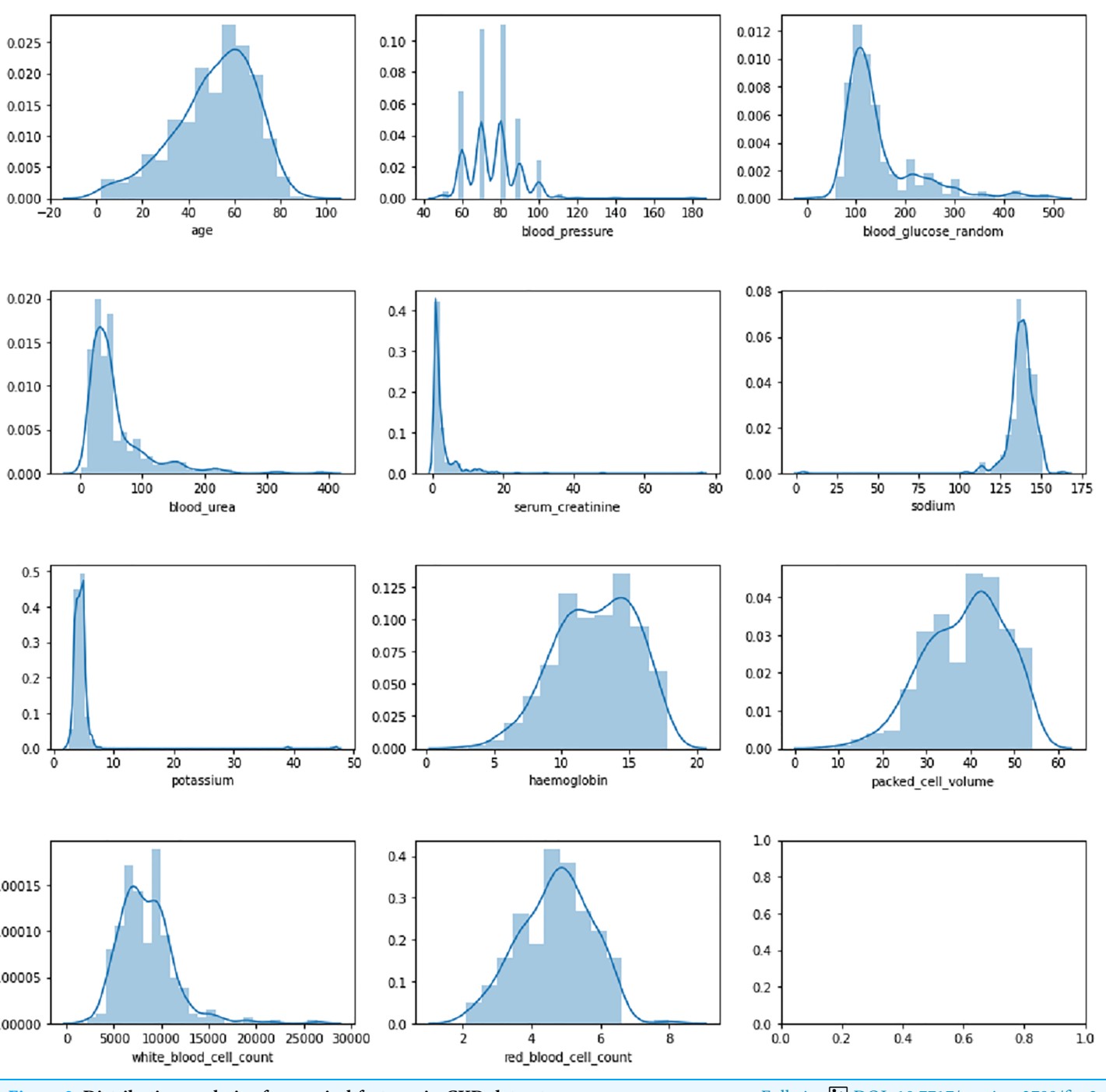

**Figure 2 Distribution analysis of numerical features in CKD dataset.**
Distributions of categorical Features

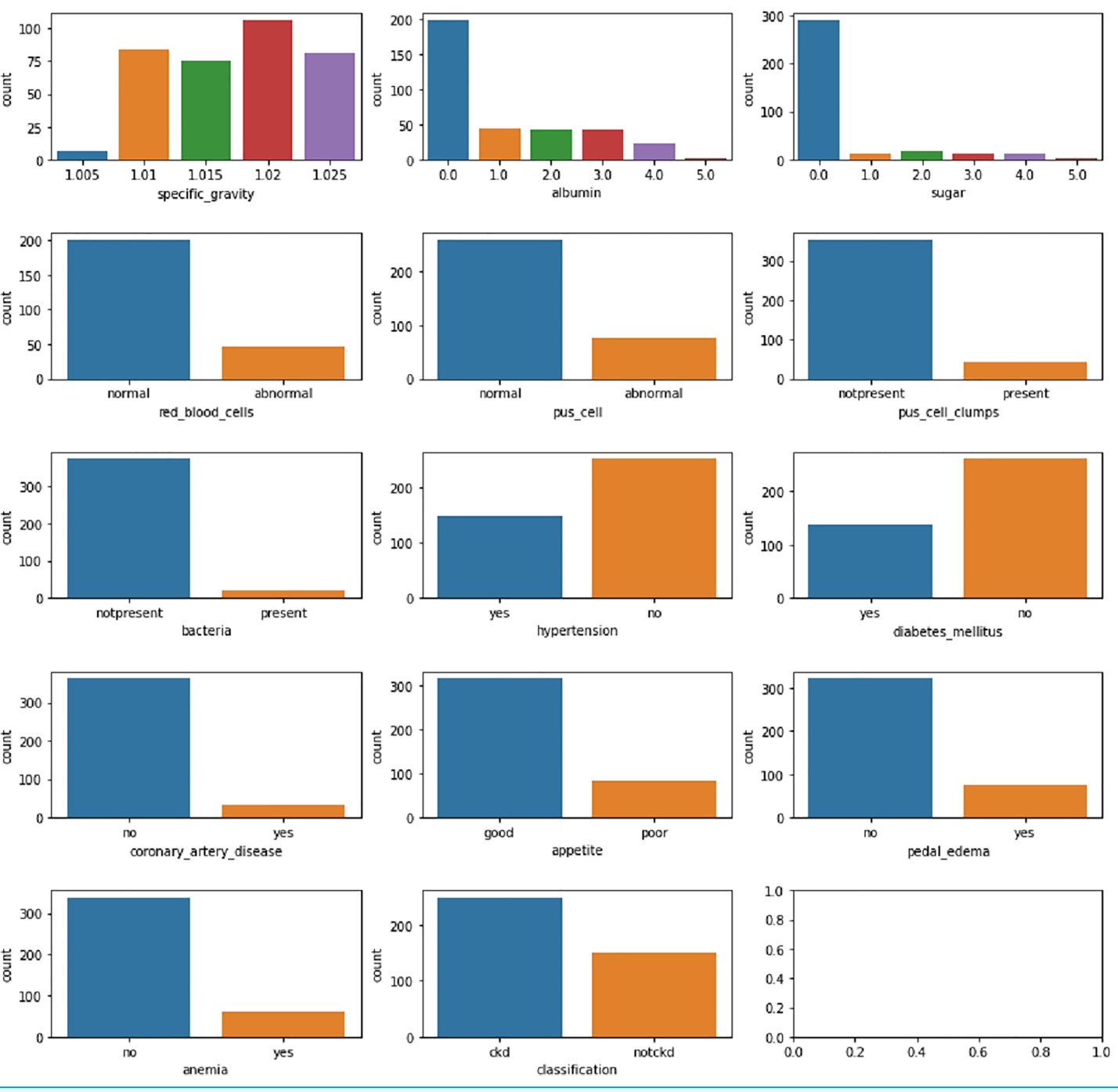

**Figure 3 Distribution analysis of categorical features in CKD dataset.**

accessed using the link given: https://archive.ics.uci.edu/dataset/336/chronic+kidney +disease.

## Experimental setup

Experiments were carried out using an Intel (R) Core (TM) i7-113H processor, which processes at 3.30 GHz and comprises 4 Core(s). The system's Random Access Memory is 16 GB, with a graphics processing unit of 4 GB. The latest version of Python, 3.12.3, is used to implement the algorithms. TensorFlow and Keras frameworks were used to implement the baseline classifier algorithms.

## Experimental evaluation

The VQC outperforms traditional machine learning and deep learning models for CKD detection. It shows higher accuracy, precision, recall, specificity, F1-score, and AUC-ROC compared to methods like support vector machine (SVM), Random Forest, and multilayer perceptron (MLP). After feature selection, VQC achieves 99.2% accuracy, further improving its performance. VQC's dynamic parameter adjustment and reduced overfitting offer greater efficiency and effectiveness. This makes it a superior model for CKD detection.

## Data augmentation description

Due to the short size of the dataset, data augmentation was done using Sparse Autoencoders (SAEs) to create synthetic samples while maintaining the original dataset's statistical distribution. By doing this, bias against majority classes was avoided and the class balance was improved. Additionally, it increased feature variety, which prevented the model from overfitting to a small number of cases and enabled it to learn broadly applicable patterns. Additionally, it helped reduce sparsity, guaranteeing that the model had enough training data for every severity level of CKD. Furthermore, cross-validation methods were used to ensure the model was assessed across several data partitions, which stopped it from picking up patterns unique to a certain dataset.

To ensure that the artificial samples produced by the SAE do not overburden the original dataset or add needless redundancy, the degree of augmentation was established empirically. The degree of imbalance in the dataset was taken into consideration when adjusting the ratio of augmented samples to original samples. In order to bring the minority class up to par with the majority class—that is, to match the largest class proportionately without unduly inflating the dataset—augmentation was done. As augmentation levels changed, performance changes were tracked using a validation set.

To encourage sparsity and ensure that only the most important features were kept in the latent space, the SAE model incorporated L1 regularization, which decreased the possibility of overfitting to small fluctuations. Dropout layers, which randomly deactivate neurons to enhance generalization, were used within the training procedure to stop the model from learning patterns from synthetic data. In order to verify that the model was effectively generalizing to unseen CKD cases rather than merely fitting to the augmented data, the impact of augmentation was evaluated using a different holdout test set. Rather

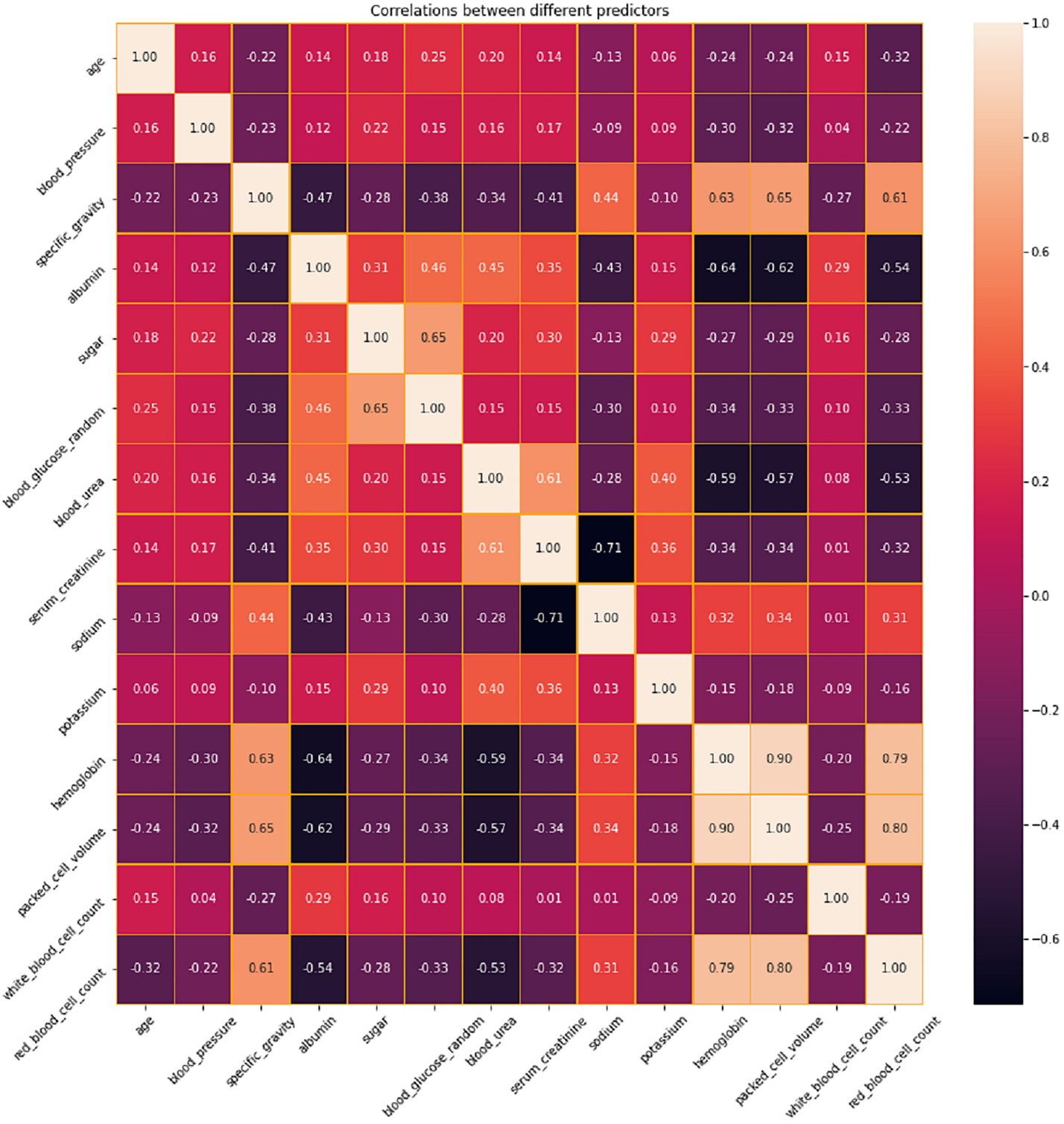

**Figure 4** Correlation analysis of features in CKD dataset.

**Table 3 Selected features using LASSO shrinkage.**

| S. No. | Feature variable | Feature description | Variable type | Value range |
|---|---|---|---|---|
| 1 | age | Age | Numerical | 0–90 |
| 2 | bp | Blood Pressure | Numerical | 0–180 |
| 3 | al | Albumin | Numerical | 0–5 |
| 4 | su | Sugar | Numerical | 0–5 |
| 5 | ba | Bacteria | Categorical | Present/Not Present |
| 6 | bu | Blood Urea | Numerical | 0–391 |
| 7 | sc | Serum Creatinine | Numerical | 0–76 |
| 8 | sod | Sodium | Numerical | 0–163 |
| 9 | pot | Potassium | Numerical | 0–47 |
| 10 | hemo | Haemoglobin | Numerical | 0–17.8 |
| 11 | pcv | Packed Cell Volume | Numerical | 0–54 |
| 12 | wbcc | White Blood Cell Count | Numerical | 0–26,400 |
| 13 | rbcc | Red Blood Cell Count | Numerical | 0–8 |

than creating an excessive amount of synthetic data at random, the final augmentation ratio was chosen based on actual performance.

## Correlation analysis

The correlation between different variables in the dataset is analyzed and presented in Fig. 4. Based on this analysis, the features with positive and negative correlations are identified. Positive correlations imply that there is a tendency for both variables to rise as one increases as well. These pairs may correspond to comparable foundational occurrences or be linked variables. When two variables have negative correlations, one variable declines as the other grows. These correlations imply that these variables have a significant inverse association. In a predictive model, highly correlated variables—whether positive or negative—might be superfluous. Removing one of the variables in a strongly correlated pair should be considered during feature selection to prevent multicollinearity, which can destabilize models and complicate the determination of coefficients. Conversely, variables that have little correlation to one another can offer distinct insights and could be more useful in a predictive model. In order to address the issues related to multicollinearity, LASSO Shrinkage is employed in this proposed system.

## Feature selection description

After the correlation analysis, feature selection is performed using LASSO shrinkage to identify the most contributing features for making efficient predictions. The features selected by the LASSO shrinkage technique are presented in Table 3. Thirteen features such as age, blood pressure, albumin, sugar, bacteria, blood urea, serum creatinine, sodium, potassium, haemoglobin, packed cell volume, white blood cell count, red blood cell count are used for the further assessment of the proposed techniques and for comparing the

results with the conventional methods or existing techniques. By picking one characteristic from a set of associated features and shrinking the rest to zero, LASSO aids in the management of multicollinearity. This enhances the comprehension of the model and lowers the chance of overfitting. LASSO helps guarantee that the model is more resilient and better generalises to new, unknown information by minimising multicollinearity. A model with less, more significant variables is easier for physicians to comprehend and recognise when it comes to CKD prediction, which makes it easier to translate model forecasts into practical choices. LASSO helps determine the important medical variables linked to the advancement of chronic kidney disease (CKD) by keeping just the most significant predictors. Through a penalty on the magnitude of the regression coefficients, LASSO shrinkage lowers the overall variance of the model and thus helps avoid overfitting.

## VQC implementation

Once the features are extracted, it is necessary to translate classical features into quantum states. RawFeatureVector from the circuit.library in Qiskit is employed for this conversion process. This creates a quantum Hilbert space from n-dimensional classical data similar to the feature transformations used in classical machine learning models, such as the polynomial kernels in SVM. For richer representations, quantum models employ entanglement and quantum superposition. The higher-dimensional quantum space makes better separation of CKD from non-CKD data possible. The next step uses the real amplitudes ansatz to define the parameterized quantum circuit (PQC). The core of VQC is PQC, which uses quantum-transformed data to identify patterns. Three layers of entangling gates with full entanglement and three trainable rotation gates (RX, RY, and RZ) are stacked by RealAmplitudes ansatz.

By facilitating information exchange between qubits, entanglement enhances the expressivity of the model. The quantum support vector machine (QSVM) simulator is then used to train the quantum model. To guarantee stable training, the gradient-free Constrained Optimization By Linear Approximations (COBYLA) optimizer is used in conjunction with the CrossEntropy loss function.

## Performance evaluation

The performance of the proposed model is assessed in three ways: without feature selection, with feature selection and with existing techniques in the literature. Initially, a comparison is performed against conventional machine learning and deep learning techniques, excluding the feature selection process. Techniques such as SVM, Random Forest, logistic regression, multi-layer perceptron, radial basis function networks, and quantum SVM are utilised for the performance comparison. Compared to traditional machine learning models for CKD detection, the VQC exhibits notable advantages. VQC naturally takes advantage of quantum feature encoding, which enables it to achieve improved class separability without predefined kernel functions; in contrast, SVM depends on kernel tricks to transfer data into a higher-dimensional space. Because it can learn optimal feature representations dynamically instead of depending on fixed

**Table 4 Performance comparison with ML/DL techniques without feature selection.**

| Techniques | Accuracy (%) | Precision (%) | Recall (%) | F1-score (%) | AUC-ROC (%) |
|---|---|---|---|---|---|
| Support Vector Machine | 91.5 | 90.6 | 90.1 | 91.2 | 91.3 |
| Random Forest | 89.3 | 88.7 | 88.3 | 88.9 | 89.1 |
| Logistic Regression | 92.7 | 91.2 | 91.8 | 92.3 | 92.5 |
| Multi-layer Perceptron | 93.6 | 92.4 | 92.7 | 93.1 | 93.3 |
| Radial Basis Function Networks | 95.9 | 94.3 | 94.7 | 95.2 | 95.5 |
| Quantum SVM | 96.7 | 95.4 | 94.9 | 96.2 | 96.4 |
| Proposed | 98.2 | 97.2 | 97.5 | 97.9 | 97.6 |

transformations, VQC has an advantage over classical SVM. Similarly, Random Forest needs a huge ensemble of decision trees, which increases computational complexity even if it works well with imbalanced datasets and non-linear connections. On the other hand, because of its quantum state representation, VQC is more efficient regarding training data requirements and delivers greater generalisation with fewer training samples.

Because it is a linear model, logistic regression has trouble capturing the intricate non-linear relationships that define the course of chronic kidney disease. Although computationally efficient, its predictive potential in datasets with complex connections is limited by the assumption of feature independence. In contrast to linear models, VQC works in a quantum-enhanced Hilbert space and is better at capturing intricate correlations in the data. A deep learning-based method called the MLP is better at managing non-linearity, but it needs a lot of training data to generalize effectively. Furthermore, it is prone to overfitting, particularly on small datasets, whereas the variational approach of VQC effectively optimizes quantum parameters to reduce the danger of overfitting.

Though radial basis function networks are sensitive to outliers and necessitate careful basis function tuning, they provide powerful non-linear classification capabilities. On the other hand, VQC eliminates the requirement for intensive hyperparameter tuning by automatically learning the best feature mappings through quantum circuit optimization. Despite using quantum principles, the quantum support vector machine (QSVM) still uses a quantum kernel, which is still a fixed transformation even though it is better than classical kernels. However, because VQC is a completely variational model, it can dynamically modify its parameters to determine the best classification decision limits.

The outcomes of the experimental evaluation are presented in Table 4, SVM produces an accuracy of 91.5%, precision of 90.6%, recall of 90.1%, Specificity of 90.8, F1-score of 91.2% and AUC-ROC of 91.3%. Random Forest exhibits 89.3% accuracy, 88.7% precision, 88.3% recall, 88.5% specificity, 88.9% F1-score and 89.1% AUC-ROC. Logistic regression produces an accuracy, precision, recall, specificity, F1-score and AUC-ROC of 92.7%, 91.2%, 91.8%, 91.5%, 92.3% and 92.5% respectively. MLP when applied for CKD prediction shows an accuracy of 93.6%, precision of 92.4%, recall of 92.7%, specificity of

**Table 5 Performance comparison with ML/DL techniques with feature selection.**

| Techniques | Accuracy (%) | Precision (%) | Recall (%) | F1-score (%) | AUC-ROC (%) |
|---|---|---|---|---|---|
| Support Vector Machine | 92.8 | 91.2 | 91.5 | 92.4 | 92.6 |
| Random Forest | 90.7 | 89.7 | 89.3 | 90.4 | 90.5 |
| Logistic Regression | 93.9 | 92.3 | 92.8 | 93.4 | 93.7 |
| Multi-layer Perceptron | 94.8 | 93.5 | 93.8 | 94.3 | 94.5 |
| Radial Basis Function Networks | 96.3 | 95.7 | 95.1 | 95.9 | 96.1 |
| Quantum SVM | 97.8 | 96.3 | 96.9 | 97.5 | 97.6 |
| Proposed | 99.2 | 98.3 | 98.5 | 98.9 | 99.1 |

92.2%, F1-score of 93.1% and AUC-ROC of 93.3%. An accuracy of 95.9% is obtained for radial basis function networks with precision, recall, specificity, F1-score and AUC-ROC of 94.3%, 94.7%, 94.5%, 95.2% and 95.5% correspondingly. Quantum machine learning technique such as QSVM was also considered for comparison with the proposed system and it was found that the QSVM exhibited an accuracy of 96.7%, precision of 95.4%, recall of 94.9%, specificity of 95.1%, F1-score of 96.2% and AUC-ROC of 96.4%. However, the proposed system showed higher values for all evaluation metrics considered such as 98.2%, 97.2%, 97.4%, 96.3%, 97.9% and 97.6% of accuracy, precision, recall, specificity, F1-score and AUC-ROC, respectively.

### Radial basis function networks

Radial basis functions are the activation functions used by RBFNs, specialized neural networks. They have been used in medical evaluations and are especially good at dynamic classification tasks. When evaluating the ability to categorize chaotic connections in medical datasets, RBFNs are a perfect comparator to VQC since they can capture complex decision boundaries. Quantum SVM: The goal of quantum SVM, an improved version of the classical SVM, is to increase classification performance by utilizing quantum computing concepts. It is a good tool for identifying micro patterns in complicated data, including early signs of chronic kidney disease. Comparing QSVM and VQC, two quantum-inspired classifiers, enables an assessment of the advantages and disadvantages of various quantum techniques to the same classification problem.

### Assessment metrics (justification)

Accuracy is a metric that provides a brief, significant summary of the model performance. A high precision indicates that the predictor model is less likely to produce false positives, which could result in needless medical procedures and patient concerns. Precision is vital for medical evaluations, particularly for CKD. Recall is essential for the prompt identification of CKD since patients' health may suffer greatly if a diagnosis is missed due to the presence of false negatives. The F1-score provides a more balanced assessment of the model's performance by combining precision and recall, particularly in unbalanced datasets where accuracy might not accurately represent the model's underlying predictive potential. Hence, these metrics are chosen for the model's performance assessment.

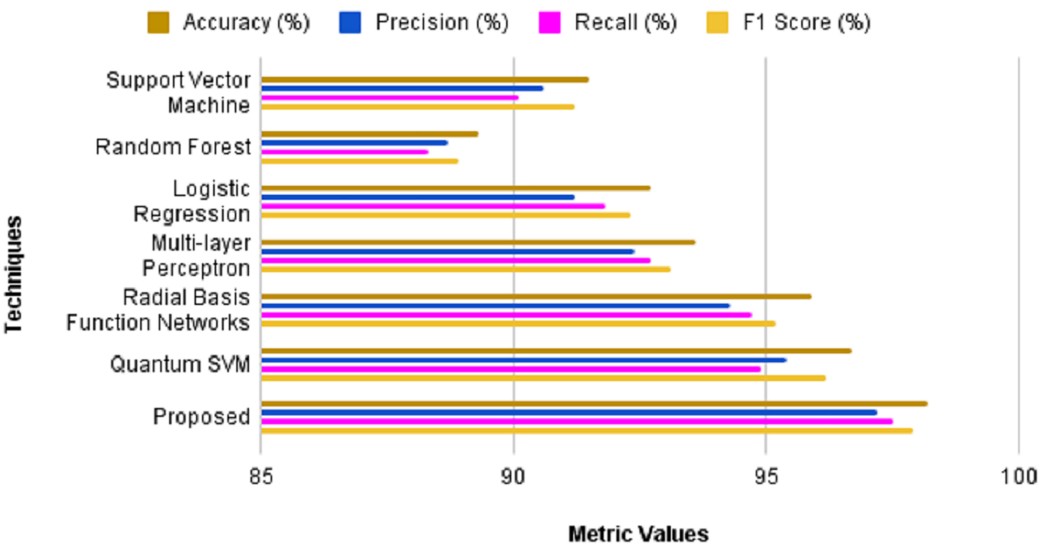

**Figure 5 Performance comparison with ML/DL methods without feature selection.**

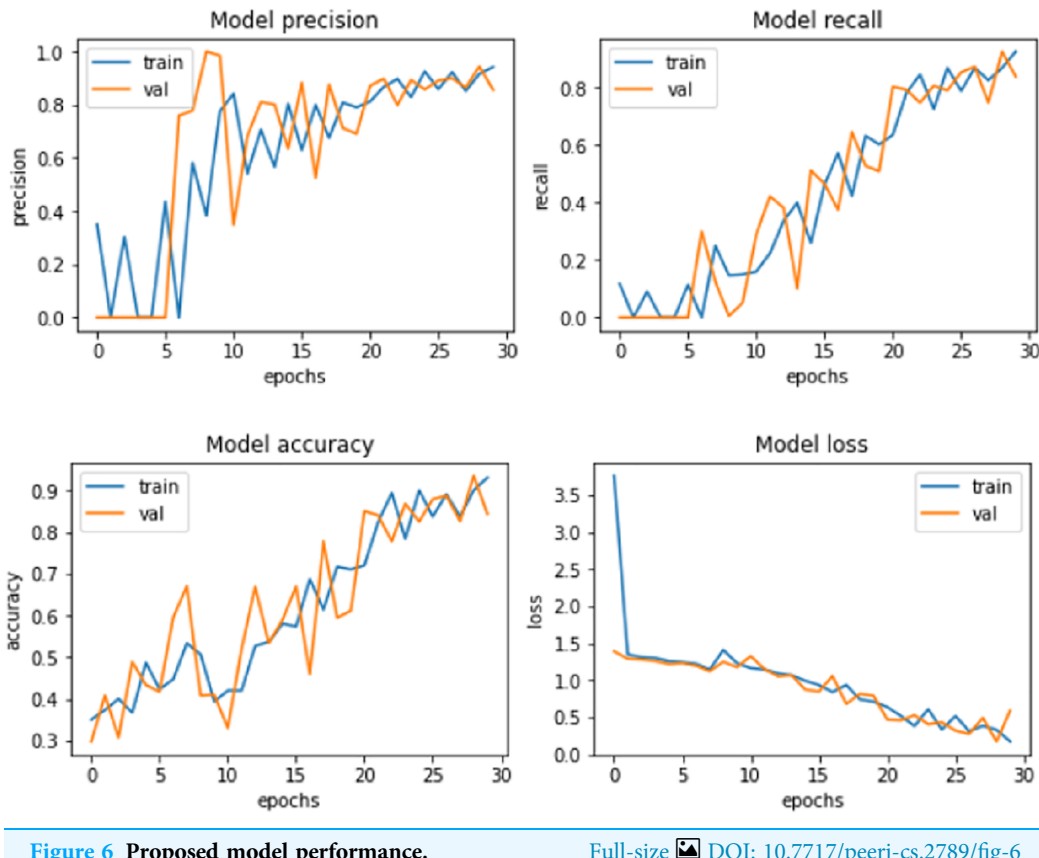

**Figure 6 Proposed model performance.**

**Table 6 Performance comparison: existing *vs* proposed.**

| References | Accuracy (%) | Precision (%) | Recall (%) | F1-score (%) |
|---|---|---|---|---|
| *Ma et al. (2020)* | 92.5 | 91.2 | 90.6 | 91.7 |
| *Peng et al. (2021)* | 94.8 | 93.2 | 92.8 | 93.7 |
| *Parab et al. (2021)* | 95.7 | 94.3 | 93.5 | 95.2 |
| *Kanda, Kanno & Katsukawa (2019)* | 97.5 | 96.5 | 95.8 | 96.3 |
| *Song et al. (2020)* | 98.4 | 97.6 | 97.8 | 98.2 |
| Proposed | 99.2 | 98.3 | 98.5 | 98.9 |

The performance of the proposed model is assessed in three ways: without feature selection, with feature selection, and with existing techniques in the literature. Initially, a comparison is performed against conventional machine learning and deep learning techniques, excluding the feature selection process. Techniques such as SVM, Random Forest, logistic regression, multi-layer perceptron, radial basis function networks, and QSVM are utilized for the performance comparison. The obtained results are presented in Table 5 and Fig. 5. The outcomes of the experimental evaluation are presented provided in Fig. 6. SVM produces an accuracy of 91.5%, precision of 90.6%, recall of 90.1% and F1-score of 91.2%. Random Forest exhibits 89.3% accuracy, 88.7% precision, 88.3% recall and 88.9% F1-score. Logistic regression produces an accuracy, precision, recall and F1-score of 92.7%, 91.2%, 91.8% and 92.3% respectively. MLP when applied for CKD prediction shows an accuracy of 93.6%, precision of 92.4%, recall of 92.7% and F1-score of 93.1%. An accuracy of 95.9% is obtained for radial basis function networks with precision, recall and F1-score of 94.3%, 94.7% and 95.2% correspondingly. Quantum machine learning technique such as QSVM was also considered for comparison with the proposed system and it was found that the QSVM exhibited an accuracy of 96.7%, precision of 95.4%, recall of 94.9% and F1-score of 96.2%. However, the proposed system showed higher values for all evaluation metrics considered such as 98.2%, 97.2%, 97.5% and 97.9% of accuracy, precision, recall and F1-score, respectively.

Similarly, the performance of the proposed system was compared against these models after the LASSO shrinkage feature selection technique was applied to the CKD dataset, and 13 significant features were identified. The models' performance showed improvements after the selected features were used to make predictions. SVM produced an accuracy of 92.8% with precision, recall and F1-score of 91.2%, 91.5% and 92.4% respectively. Random Forest method showed an increase in accuracy of 90.7%, precision of 89.7%, recall of 89.3% and F1-score of 90.4%. Logistic regression method produced an accuracy level of 93.9% in making predictions with precision, recall and F1-score of 92.3%, 92.8% and 93.4% respectively. MLP showed an accuracy of 94.8%, precision of 93.5%, recall of 93.8%, and F1-score of 94.3%. Radial basis function networks exhibited 96.3%, 95.7%, 95.1% and 95.9% of accuracy, precision, recall and F1-score accordingly. The accuracy of QSVM increased as 97.8% with precision of 96.3%, recall of 96.9% and F1-score of 97.5%. The proposed model produced an improved accuracy of 99.2%, which is greater than the

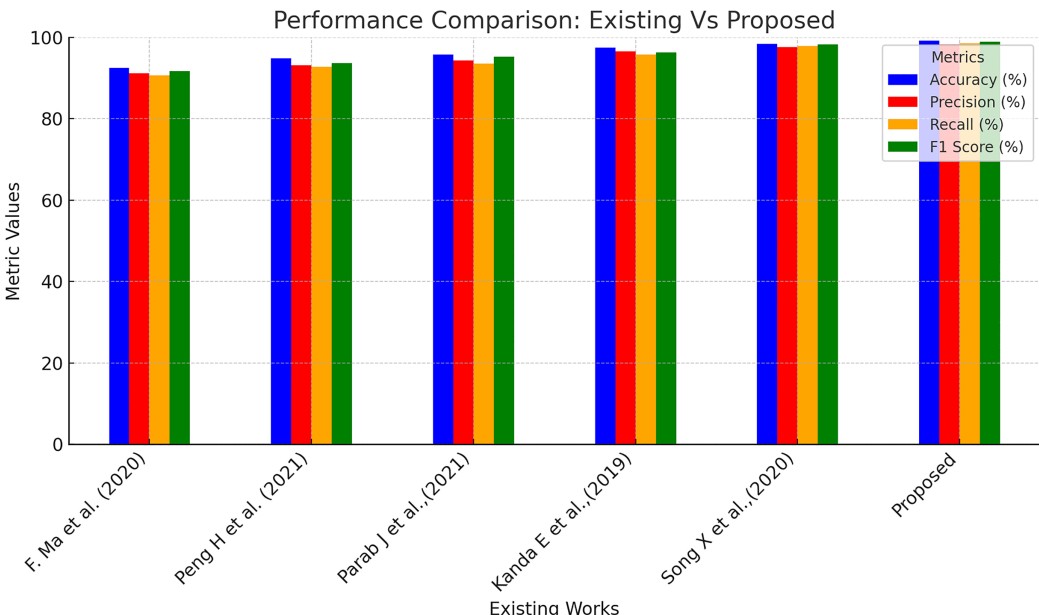

**Figure 7** **Performance comparison of existing and proposed methods.**

accuracy exhibited by the other models considered for comparison. The values of precision (98.3%), recall (98.5%), and F1-score (98.9%) were also higher for the suggested model. The model performance of the proposed system is presented in Fig. 6.

Further, the performance of the proposed model was compared with a few existing models in the literature for CKD predictions, and the outcomes are presented in Table 6 and Fig. 7. *Ma et al. (2020)* suggested a heterogeneous modified artificial neural network technique for CKD prediction. This model was 92.5% accurate in making predictions with 91.2%, 90.6%, and 91.7% of precision, recall, and F1-score correspondingly. *Peng et al. (2021)* employed a two-stage neural network for predicting kidney diseases and showed an accuracy level of 94.8%. Another work incorporated a backpropagation neural network-based machine learning model to classify 'CKD' and 'Not CKD' classes. This model produced an accuracy of 95.7% with 94.3% and 93.5% of precision and recall values, respectively. Bayesian network combined with artificial intelligence techniques were utilized in *Kanda, Kanno & Katsukawa (2019)* for CKD detection, and the accuracy level of the suggested model was found to be 97.5%. In the work proposed in *Song et al. (2020)*, a temporal-enhanced gradient boosting machine was employed for predicting kidney diseases, and this model exhibited an accuracy of 98.4%. Table 6 presents the performance of the proposed model is higher compared to the existing works taken for investigative comparison for all four assessment metrics.

## Limitations of current research

Quantum circuits are susceptible to errors because of noise sources and distortion. These errors may result in predictions that necessitate intricate error correction, which is still

underdeveloped for large-scale quantum calculations. When there are more qubits and gates in a quantum circuit, it becomes challenging to develop and optimise for VQCs. This intricacy may restrict the method's scalability, particularly in the case of complicated models needed for high-dimensional medical data. There can be a large overhead associated with mapping classical properties into quantum states regarding both time and computing efficiency. The financial burden may reduce the potential efficiency offered by quantum computing for CKD prediction.

## CONCLUSION

This study establishes the groundwork for upcoming developments in quantum-assisted medical AI systems by proving the feasibility of QML for CKD diagnosis. The proposed system efficiently manages the complexity of CKD datasets by incorporating Kalman filter and normalisation algorithms for pre-processing, sparse autoencoders for data augmentation and LASSO shrinkage for feature selection. Using VQCs has produced a robust solution for early CKD detection, surpassing standard classifiers with a high classification accuracy of 99.2%. However, while VQCs exhibit encouraging outcomes, the limitations of present quantum hardware increase the training time. The future subsequent investigations on this area ought to concentrate on expanding this structure to more extensive and varied datasets to verify the adaptability and applicability of the model. Furthermore, investigating incorporating additional quantum algorithms and hybrid classical-quantum models may considerably improve computational efficiency and performance. The suggested architecture may develop into a potent early disease detection tool as quantum computing technology advances, providing revolutionary advantages for world healthcare.

### Funding

This work was supported by the Korea Environmental Industry & Technology Institute (KEITI), with a grant funded by the Korean government, Ministry of Environment (The development of IoT-based technology for collecting and managing big data on environmental hazards and health effects), Grant RE202101551 and Princess Nourah bint Abdulrahman University Researchers Supporting Project number (PNURSP2024R237), Princess Nourah bint Abdulrahman University, Riyadh, Saudi Arabia. The funders had no role in study design, data collection and analysis, decision to publish, or preparation of the manuscript.

### Grant Disclosures

The following grant information was disclosed by the authors:
Korean Government, Ministry of Environment: RE202101551.
Princess Nourah bint Abdulrahman University Researchers Supporting Project: PNURSP2024R237.
Princess Nourah bint Abdulrahman University, Riyadh, Saudi Arabia.

## Competing Interests

The authors declare that they have no competing interests.

## Author Contributions

- P. Parthasarathi conceived and designed the experiments, performed the experiments, analyzed the data, performed the computation work, authored or reviewed drafts of the article, and approved the final draft.
- Haya Mesfer Alshahrani conceived and designed the experiments, performed the experiments, analyzed the data, performed the computation work, prepared figures and/or tables, and approved the final draft.
- K. Venkatachalam conceived and designed the experiments, performed the experiments, analyzed the data, performed the computation work, prepared figures and/or tables, and approved the final draft.
- Jaehyuk Cho conceived and designed the experiments, performed the experiments, analyzed the data, performed the computation work, authored or reviewed drafts of the article, and approved the final draft.

## Data Availability

The kidney disease dataset is available at Zenodo: Parthasarathi, P. (2024). Variational Quantum Classifier-based early identification and classification of chronic kidney disease using Sparse autoencoder and LASSO Shrinkage [Data set]. Zenodo. https://doi.org/10.5281/zenodo.13751242.

The Chronic Kidney Disease dataset is available at UCI Machine Learning Repository: Rubini, L., Soundarapandian, P., & Eswaran, P. (2015). Chronic Kidney Disease [Dataset]. UCI Machine Learning Repository. https://doi.org/10.24432/C5G020.

## Supplemental Information

Supplemental information for this article can be found online at http://dx.doi.org/10.7717/peerj-cs.2789#supplemental-information.

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
