# Peer review of "Variational quantum classifier-based early identification and classification of chronic kidney disease using sparse autoencoder and LASSO shrinkage"

_PeerJ Computer Science, doi:10.7717/peerj-cs.2789_

## Round 0.1 · original submission · Major Revisions

Dear Authors,

Your paper has been reviewed. Based on the reviewers' reports, major revisions are needed before it is considered for publication in PEERJ Computer Science. The issues you have to fix in your revised version of your paper are mainly the following:

1) You must include other key performance metrics to evaluate the classifier's performance comprehensively. Furthermore, you must demonstrate the role of Lasso shrinkage in selecting features and provide a clear list of selected features and their importance.

3) The Variational Quantum Classifier (VQC) implementation needs to be detailed. Because you claim the superior performance of your method, comparisons with other standard classifiers (e.g., SVM, Random Forest, or XGBoost) are required.

3) You must improve the dataset processing section by providing a more detailed explanation of data augmentation.

Reviewer 1 ·

Basic reporting

1. While the accuracy (92.9%) is reported, other key performance metrics like sensitivity, specificity, and AUC-ROC should be included to provide a more comprehensive evaluation of the classifier's performance.
2. Sparse autoencoders are used for data augmentation to balance the smaller classes, but the authors do not quantify or justify the level of augmentation. Overfitting risks are not addressed.
3. The Kalman filter is used for noise reduction without specifying the noise model assumptions or providing evidence of its suitability for CKD datasets.
4. The paper uses the UCI CKD dataset, which is relatively small (400 instances) and lacks real-world variability. The generalizability of the model to larger or more diverse datasets is questionable.
5. The role of Lasso shrinkage in selecting features is stated but not demonstrated. A clear list of selected features and their importance should be provided.
6. The implementation of the Variational Quantum Classifier (VQC) is not detailed, and the quantum advantage over classical ML models is not convincingly shown.
7. While the paper claims superior performance, comparisons with other standard classifiers (e.g., SVM, Random Forest, or XGBoost) are missing, which weakens the novelty claim.
8. The authors can address methodological gaps and enhance the rigor of their study by refer some state of the art methods, https://doi.org/10.1016/j.compbiomed.2023.106646, https://doi.org/10.1109/iSAI-NLP51646.2020.9376787, https://doi.org/10.1109/ACCESS.2021.3053759.

Experimental design

please see the comments of basic report, section 1

Validity of the findings

please see the comments of basic report, section 1

Additional comments

please see the comments of basic report, section 1

·

Basic reporting

The mortality rate from chronic kidney disease is high. Therefore, early detection of the disease is of great importance. To classify this disease, the authors used modern deep machine learning techniques. Thus, this article demonstrates scientific novelty and practical significance.
1. Your abstract requires more detailed information. I suggest expanding the abbreviations in lines 22–32.
2. I thank the authors for the detailed literature review. Additionally, the manuscript is clearly and professionally written with unambiguous language. I would advise the authors to include a problem statement after the "Related Works" section.

Experimental design

1.The methods described in the article are sufficiently detailed. However, the article would be more comprehensible if a generalized algorithm for applying these methods were presented after their description.
2. I appreciate the description of dataset processing. Although your results are convincing, the dataset processing section could be improved by providing a more detailed explanation of data augmentation.

Validity of the findings

The authors conducted sufficient experiments and evaluations. The article would be enhanced if the conclusion section detailed the scientific novelty and practical significance of this research.

Additional comments

1. The authors used a Variational Quantum Classifier. However, quantum-based classifiers are currently slow in training.
2. The authors used pre-existing tools such as Sparse Autoencoder and Lasso Shrinkage. However, no enhancements or improvements to these methods were performed in the study.

---

## Round 0.2 · accepted · Accept

Dear Authors,
Your paper has been revised. It has been accepted for publication in PEERJ Computer Science. Thank you for your fine contribution.

Reviewer 1 ·

Basic reporting

Authors have addressed all of the comments adequately. I have no further comments on this paper.

Experimental design

no comment

Validity of the findings

no comment

Additional comments

no comment